# Place of Residence and Marital Status as Variables Differentiating a Sense of Self-Efficacy in the Elderly—A Descriptive Cross-Sectional Survey

**DOI:** 10.3390/healthcare8030300

**Published:** 2020-08-26

**Authors:** Klaudia Jakubowska, Mariusz Wysokiński, Paweł Chruściel

**Affiliations:** Department of Basic Nursing and Medical Teaching, Chair of Development in Nursing, Faculty of Health Sciences, Medical University of Lublin, 20-081 Lublin, Poland; mariusz.wysokinski@umlub.pl (M.W.); pawelchrusciel@o2.pl (P.C.)

**Keywords:** old age, senior age, effectiveness, sense of efficacy, health behaviors

## Abstract

*Background:* Studies on the self-efficacy of the elderly are still being conducted to a very limited extent. Nevertheless, they can provide relevant information for specialists to use in many disciplines of science, as well as for practitioners, especially gerontologists, geriatricians, psychologists and other people associated with prophylaxis and health promotion. The aim of the study is to assess the impact of the place of residence and marital status on the self-efficacy in the elderly. *Methods:* A diagnostic survey method was used for the study and within the method, the authors’ sociodemographic data questionnaire and the generalized self-efficacy scale (GSES) were applied. The study was carried out in a group of 171 females and 130 males under nursing care in the home environment or in nursing-home care in Poland. *Results:* The average level of generalized sense of self-efficacy in the research cohort was M = 26.69, SD = 8.49. A relatively high percentage of the group (48.9%) showed a sense of self-efficacy at a high level. The level of a sense of self-efficacy was low in 38.2% of the cohort, while its average intensity in the remaining ones constituted 12.9% of the total. A statistically significant impact of a place of residence (Z = −2.940; *p* = 0.003) and marital status (H = 12.000; *p* = 0.007) on the sense of efficacy in the research cohort was proven in favor of those from the rural environment and the respondents having spouses. *Conclusions:* On the scale of the studied group, the results of a sense of self-efficacy are optimistic, as older people with high levels predominated. However, the results of the cohort whose self-efficacy is insufficient to cope with their old age should be considered carefully, especially in the case of lonely people living in the urban environment and NHC (nursing-home care) residents.

## 1. Introduction

We have been living longer than ever before, and the age profile for the whole society is changing dynamically. The number of people aged 65 or older will increase by more than 40% within 20 years, whereas the number of households in which the most senior dweller is 85 or older is growing faster than the number of households with oldest inhabitants from any other age group. These changes bear far-reaching consequences, hence increasing studies focus on the elderly age group [1].

Aging is an individualized process and should be analyzed from many perspectives: from a biologic through psychological up to a social one. The beginning of the old age is usually connoted with the retirement age, which is reflected in demographic and sociological studies. The age of 60 or 65 is considered to be the threshold for both males and females [2,3].

Entering the old age stage, which is usually associated with retirement, is one of the watershed moments in everyone’s life and is mistakenly considered in terms of the so-called difficult and critical situations. It is connected with a loss of the current professional and material status, diminishing one’s sense of significance and prestige and thus intensifying a sense of uselessness [1,4,5].

The concept of efficacy is an ambiguous term that is commonly combined with concepts of agency, resourcefulness and entrepreneurship. It is currently included in personal resources, which are available to every human being to a higher or lower degree. A sense of self-efficacy allows one to anticipate intentions and actions in various areas of activity—including health behaviors—as it becomes an implication of currently undertaken actions and the intention to engage in health activities [6,7].

A sense of self-efficacy can be understood as a kind of psychological mechanism conditioning a change in behaviors and attitudes—as well as the effectiveness of actions taken. It constitutes an intermediary link between one’s knowledge and behavior. It has the form of the belief that one is able to effectively conduct a given action to achieve the expected result [8].

Analysis of the available literature indicates rather poor empirical insight into the phenomenon of the sense of efficacy in elderly people. Studies that have been conducted in this age group to date have been focusing worldwide on the general evidence that enjoying high self-efficacy may be favorable for the aging process [9,10]. A sense of self-efficacy has also been juxtaposed with other variables such as health behaviors, social support and social participation [11]. However, literature review has failed to reveal any studies in which sociodemographic variables such as the place of residence and marital status would be exposed. Research with other sociodemographic variables such as age, sex, social status, the number of inhabitants in a household has been conducted both in Poland and abroad. Nonetheless, these need to be supplemented and updated with research involving other sociodemographic variables [6,7,12,13].

Within Poland seen as a fragment of Europe, there is a constant need to conduct and streamline research into factors affecting improvements in the care of the elderly people. Thus, this research has been undertaken in order to facilitate taking care of the elderly and to expand details of current general findings and to focus on how sociodemographic variables such as the marital status and the place of residence may specifically differentiate the sense of self-efficacy. This investigation provides new evidence suggesting urban dwellers and lonely people suffer from a decreased self-efficacy. This research suggests that programs aimed at taking care of the elderly ought to be drafted in the light of the research findings as this may be significant for the cooperation with a patient.

## 2. Materials and Methods

The study was based on a quantitative strategy—a cross-sectional survey method. The study was conducted in Poland from February to September 2017. The selection of the group was based on purposeful sampling within the scheme based on the availability of respondents. The research cohort consisted of 301 respondents (171 females and 130 males) under nursing care in the home environment and daytime nursing-home care teams in Poland. Staying in a nursing-home care or being under nursing care in one’s home environment, being above 65 and a result of the mini-mental state examination (MMSE) where the cutting point was 27 points, were all the criteria for being included into the research. For diagnostic purposes, the sociodemographic data questionnaire and adjusted Polish-language generalized self-efficacy scale (GSES) were used. The questionnaire allowed collecting data characterizing the research cohort in terms of selected variables. The generalized self-efficacy scale—GSES by R. Schwarzer, M. Jerusalem was used to measure the generalized sense of self-efficacy. It measures the validity of the general belief of the people regarding the effectiveness of coping with difficult situations and obstacles. It consists of 10 statements, where a respondent can choose between 4 answers to each question: no—1; rather not—2; rather yes—3; yes—4. The sum of all the results amounts to a general sense of self-efficacy indicator, which ranges between 10 and 40 points. It is assumed that the higher the score, the greater the sense of self-efficacy. The tool has sten standards, a high reliability factor (Cronbach’s alpha = 0.96).

The research was approved by the Bioethics Committee of the Medical University of Lublin in Poland [no. KE-0254/91/2017]. The respondents were informed about the scientific purpose of collecting data and assured of anonymity. The empirical material was developed using the IBM SPSS Statistics (version 21, IBM, Armonk, NY, USA) statistical package. In order to check the impact of the place of residence on respondents’ sense of efficacy, the Mann–Whitney test was used. In turn, the Kruskal–Wallis test checked the impact of respondents’ marital status on their sense of efficacy. The adopted level of statistical significance was *p* < 0.05. The results meeting the criterion of statistical significance were marked in bold.

## 3. Results

The respondents’ average age was 76 (M = 76.70; SD = 7.05). The youngest respondent was 65 and the oldest was 95 years old. Table 1 features detailed sociodemographic data.

On the GSES scale, the average result for respondent generalized self-efficacy was M = 26.69, SD = 8.49. Of the respondents, 48.9% gained a high score for their sense of efficacy. A low score was recorded in 38.2% of cases, while an average score was recorded in 12.9% of the total. The data converted to sten standards are presented in Table 2.

The respondents most strongly agreed with the statement “*If someone opposes me, I know the ways to achieve what I want*” (M = 2.75; SD = 1.01), while they agreed least with the statement “*Thanks to my ingenuity I can cope with unexpected situations*” (M = 2.62; SD = 1.01). A detailed distribution of respondents’ answers the GSES statements is presented in Table 3.

Studies have confirmed that in the case of the research cohort, the place of residence differentiates a sense of self-efficacy in the elderly. The respondents living in the country obtained higher scores on the scale of a sense of self-efficacy than respondents living in the city (Z = −2.940; *p* = 0.003) (Table 4).

Studies have shown that marital status differentiates a sense of efficacy in elderly people (H = 12.000; *p* = 0.007). The highest average score was recorded in married respondents and the ones living with their families. A lower score was obtained by lonely (unmarried, widowed, divorced) respondents, including the ones staying in the NHC (Table 5).

## 4. Discussion

Demographic forecasts of the population until 2050 clearly indicate the aging in the Polish society. This trend is of a global nature. There is a tendency to reduce the number of births and, at the same time, to extend lifespans in favor of women. While in 2000 there were about 600 million people in the world aged 60 and more, in 2025 they are expected to reach the number of about 1.2 billion miliardand in 2050 the number of two trillion seniors. The number of elderly people in developing countries is steadily increasing [14]. The studies assumed that there was a direct cause and effect relationship between the psychological mechanism of a sense of self-efficacy and behavior. It was recognized that understanding a sense of self-efficacy in elderly people can contribute to a better understanding of their capabilities, limitations and needs [6]. Numerous studies having been conducted in recent years among students at the University of the Third Age (UTA) confirm that, compared to their peers who do not show such activity, active UTA students have a higher level of confidence related to the chances of achieving their goals. They define their goals more easily, cope with stress and health problems far better and achieve higher results when implementing changes in pre-existing behaviors that may be harmful to their health [15,16]. Physical activity is connected with a high indicator of one’s sense of self-efficacy. The functioning of the elderly surveyed was also significantly influenced by their family or a spouse. Elderly people who were active in their lives were characterized by: efficiency, task-oriented approach to problems, a firm sense of meaning in their lives, self-esteem, a belief in success, even in environments that provide merely limited opportunities [17,18].

A place of residence, as a sociodemographic variable, may be highly significant for specifying patient’s personal resources. Diabetic patients living in rural areas have been shown to lack sufficient knowledge on their disease, which was connected with the specific nature of living in the country. Furthermore, factors such as lack of time and opportunities to broaden their knowledge decreased respondents’ assessments of their self-efficacy [19]. In addition, the presented studies confirmed a statistically significant impact of the place of residence on the sense of efficacy in the elderly in favor of the rural environment. This regularity is probably connected with the specificity of life in the country, which entails a more active lifestyle resulting from the number and the load of responsibilities in the household. This corresponds to Japanese research according to which a sense of self-efficacy in elderly rural residents is more outlined than among the city dwellers. What is more, the higher it is, the firmer the tendency to undertake health behaviors and maintain global life activity [20]. This regularity requires further analysis, especially if one considers the expectation that elderly people from the city, as more aware, often better educated, having easier access to health institutions, culture and even education of seniors, constitute the group with a better perception of their self-efficacy.

In addition, the authors’ own research has shown that marital status has a statistically significant effect on a sense of self-efficacy in elderly respondents in favor of those having spouses. Having a spouse and being close to each other can mean not only a sense of security for a senior, but also greater mobilization and faith in their own abilities. International literature confirms that the general life situation of the elderly—as well as their fitness—to be, in fact, connected with their sense of self-efficacy—which largely depends on their marital status. [21]. Focusing on the marital status as a sociodemographic variable is especially significant in the Polish context. National Statistics Poland shows males to have a higher death rate. Over 75% of males aged 65 or older are married, whereas 58% of females in this age group are widows [22]. The authors’ own research findings show loneliness to weaken the sense that one is able to undertake certain actions or meet the goals they set. Furthermore, it negatively affects the assessment of an individual’s personal resources. This finding may prove extremely useful for nursing practices. Being close to one’s spouse may be believed to constitute not only a sense of security, but also greater resources for mobilizing and faith in one’s abilities, which is especially significant at the old age when one may be troubled by various ailments. [23] This premise was confirmed by one of the European research studies by Skiba and Duda et al. [24]. The authors showed that a sense of self-efficacy helped to cope more effectively with the challenge involving cancer. Thus, they confirmed the positive correlation between self-efficacy and health behaviors. The elderly and chronically ill with a high sense of self-efficacy will be more likely to undertake health-promoting behaviors and will be more determined to maintain them. This is not insignificant in the case of elderly people, because they are often affected by various diseases, including cancer. Hence, a sense of self-efficacy has been shown to be interconnected with numerous variables that directly or indirectly affect one’s health state. Marital status is an important variable which medical stuff checks at the very beginning of an interview with their patient. Being aware how marital status may influence the level of the sense of self-efficacy, the medical personnel may forecast possible deficits at the very beginning of their interview and provisionally plan individual stages of the cooperation [25].

### Study Limitations

The presented study is not free from limitations, with the first being the fact that the cohorts of urban and rural dwellers were unequal. This shortcoming needs to be addressed while planning further research. In addition, the research involved elderly patients requiring a greater involvement from the researcher while collecting the study material. Some respondents live permanently in the nursing-home care, which may be the criterion for differentiating the obtained findings. In addition, a more multifaceted perspective of the self-efficacy at an old age would apply a broader range of diagnostic tools to a larger population of seniors from various regions of Poland. Future investigations need to be supplemented with other variables that may differentiate levels of the sense of self-efficacy in the elderly.

## 5. Conclusions

The level of the sense of self-efficacy ensuring comfort and acceptance of one’s old age is more characteristic of elderly people living in the country and married ones—not being under the care of the NHC. It is necessary to review activities undertaken in the area of social policy in Poland, related to the care in social welfare homes. It is important to undertake actions aimed at increasing the level of self-efficacy in Polish seniors, as well as to use available social and financial resources of the government and social associations for this purpose. Health promotion programs need to be drafted that could increase or improve levels of the sense of self-efficacy in elderly people living alone in urban areas. A sense of self-efficacy is hope for the elderly in psychological, social and health dimensions. Owing to this, they can be more active, open, responsible for their attitudes and behavior and thus fit into the modern pattern of an old age. Owing to a positive perception of self-efficacy, it is possible for senior citizens to shape a more aware and active sense of responsibility for their health. This may also help them undertake more pro-health behaviors.

## Figures and Tables

**Table 1 healthcare-08-00300-t001:** Respondents’ sociodemographic data.

Sociodemographic Variables	*n*	%
Gender		
Female	171	56.8
Male	130	43.2
Age		
Up to 70 years old	72	23.9
71–75 years old	62	20.6
76–80 years old	83	27.6
Over 80	84	27.9
Place of residence		
City	252	83.7
Rural area	49	16.3
Marital status		
Bachelor/maiden	31	10.3
Married	82	27.2
Divorced	39	13.0
Widowed	149	49.5
Place of stay		
Family home	15	51.8
Nursing home	145	48.2

**Table 2 healthcare-08-00300-t002:** Distribution of sten scores examined in the generalized self-efficacy scale (GSES).

Stens	*n*	%
1—low score	54	17.9
2—low score	13	4.3
3—low score	24	8.0
4—low score	24	8.0
Total of low score	115	38.2
5—average score	13	4.3
6—average score	26	8.6
Total of average score	39	12.9
7—high score	67	22.3
8—high score	30	10.0
9—high score	22	7.3
10—high score	28	9.3
Total of high score	147	48.9

**Table 3 healthcare-08-00300-t003:** Distribution of respondents’ answers to individual statements of the GSES.

Statement’s Content	*n*	Min	Max	M	SD
I can always solve difficult problems if I try hard enough	301	1.00	4.00	2.68	0.94
If someone opposes me, I know ways to achieve what I want	301	1.00	4.00	2.75	1.01
It is easy for me to stick to my goals and achieve them	301	1.00	4.00	2.67	1.05
I am convinced that I could deal effectively with unexpected events	301	1.00	4.00	2.63	0.99
Thanks to my ingenuity, I can cope with unexpected situations	301	1.00	4.00	2.62	1.01
I can solve most problems if I put enough effort in it	301	1.00	4.00	2.67	0.97
I can remain calm when facing difficulties because I can rely on my coping skills	301	1.00	4.00	2.68	0.90
While I am struggling with a problem, I can usually find several solutions	301	1.00	4.00	2.71	0.93
When I am in an awkward situation, I generally know what to do	301	1.00	4.00	2.66	0.90
Regardless of what happens to me, I can deal with it	301	1.00	4.00	2.63	0.98
Sense Of Self-Efficacy (SSE)	301	10.00	40.00	26.69	8.49

**Table 4 healthcare-08-00300-t004:** Place of residence and the level of self-efficacy in the respondents.

Demographic Variables		Sense of Self-Efficacy			Mann–Whitney Test	
		*n*	M	SD	Z	*p*
Place of Residence	*City*	252	26.05	8.54	−2.940	0.003
	*Country*	49	30.00	7.44		

**Table 5 healthcare-08-00300-t005:** Marital status and the level of a sense of self-efficacy in the respondents.

Demographic Variables		Sense of Self-Efficacy			Kruskal–Wallis Test	
		*n*	M	SD	H	*p*
**Marital Status**	*Single male/female*	31	22.35	8.57	12.000	0.007
	*Married male/female*	82	29.26	6.34		
	*Divorced male/female*	39	26.28	9.12		
	*Widowed male/female*	14	26.29	8.94		

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
