# Peer review of "Place of Residence and Marital Status as Variables Differentiating a Sense of Self-Efficacy in the Elderly—A Descriptive Cross-Sectional Survey"

_healthcare, 2020, doi:10.3390/healthcare8030300_

Round 1
Reviewer 1 Report
Thank you for giving me to review your manuscript. The manuscript is well written and scientifically significant. This article can contribute to reconsiderations of the care of older people in communities and nursing homes regarding self-efficacy.
This study's abstract describes the contents of the primary document comprehensively, showing the importance of self-efficacy for older people. It can include the suggestion for the following research.
The background section describes the present conditions of the self-efficacy of older people in communities. My question is, "what is your research question?" The precise research question can describe your research's purpose and persuade readers to read your research. It should be clarified.
In the method section, the authors comprehensively describe the contents of the study methods. My suggestions are the clarification of the sampling method. The authors stated purposeful random sampling. What does it mean? Sampling is essential for the external and internal validity of studies. This should be clarified. Besides, is the questionnaire assessing the sense of self-efficacy translated into the authors' endemic language? Or were they used in the English version? This point is critical for the validity and reliability of this research.
In the result section, the authors clearly described the findings.
In the discussion section, the authors comprehensively described the learning points from this research. One suggestion is the congregation of the paragraphs. There are many paragraphs readability. Several paragraphs should be blended when it comes to readability. The authors can add implications for the following research to improve older people in long-term care facilities.
Author Response
Thank you very much for the reviews. My answer is in the file.

Reviewer 2 Report
I appreciate the work done by the authors in the development of the study and in the realization of the article.
No reference is made to other variables and aspects related to self-efficacy that are discussed. The introduction is poor.
It is more appropriate that the objective appears at the end of the introduction, not in materials and methods
Materials and methods should indicate the type of study, and include the criteria for inclusion and exclusion of participants.
The MMSE does not say the cut-off point, nor does it include other screening tests that it would be important to have considered a geriatric depression scale.
There ius a lack of information on the sociodemographic data of the sample. The size of the sample according to marital status is not known (only that the results of the GSES scale in Table 4 are widows and samples) but the "n" of each group is not known.
It´s stated that the sense of effectiveness is greater in people living in rural areas, 16.3%, but the percentage of participants is very small compared to the group living in urban areas.
In the discussion, the relationship between self-efficacy and other variables is discussed. But the authors' study does not study whether there is a relationship with other important variables in ageing, for example, lifestyle, life satisfaction or optimism.
The social support network is also mentioned as a factor, but no social support scale was used, only marital status, living alone or not, was taken into account.
It´s only related to sociodemographic variables such as marital status or place of residence, but nothing related to other social or psychological variables, which makes it a poor study.
Bibliographic references aren´t very current, for example, some important ones such as 12 and 13 are from 2016. Check in the bibliography the numbers in red and the underlined ones in yellow 14, 15, 19 I have marks.
Author Response

(The authors gave the same response as above.)

Round 2
Reviewer 2 Report
I appreciate the modifications made by the authors and the work to improve the article.
I appreciate the changes made, and I appreciate the work of rewriting much of the article, considering the aspects pointed out in the first review.
There is an improvement in the quality of the article, although I do not see the % of participants from the city and the country as positive in the methodology, needing to increase the number of participants from the country.